# Doctors’ Extended Shifts as Risk to Practitioner and Patient: South Africa as a Case Study

**DOI:** 10.3390/ijerph17165853

**Published:** 2020-08-12

**Authors:** Koot Kotze, Helene-Mari van der Westhuizen, Eldi van Loggerenberg, Farah Jawitz, Rodney Ehrlich

**Affiliations:** 1Nuffield Department of Primary Care Health Sciences, Radcliffe Primary Care Building, University of Oxford, Oxford OX2 6GG, UK; helene.vanderwesthuizen@phc.ox.ac.uk; 2Helen Joseph Hospital, Rossmore, Johannesburg 2092, South Africa; eldivanloggerenberg@gmail.com; 3Saïd Business School, University of Oxford, Oxford OX1 1HP, UK; farahjawitz@gmail.com; 4Division of Occupational Medicine, School of Public Health and Family Medicine, University of Cape Town, Observatory, Cape Town 8001, South Africa; rodney.ehrlich@uct.ac.za

**Keywords:** shift work, extended shifts, on-call, working hour organization, duty hours, fatigue, occupational hazards, sleep deprivation

## Abstract

Extended shifts are common in medical practice. This is when doctors are required to work continuously for more than 16 h, with little or no rest, often without a maximum limit. These shifts have been a part of medical practice for more than a century. Research on the impact of fatigue presents compelling evidence that extended shifts increase the risk of harm to patients and practitioners. However, where the number of doctors is limited and their workloads are not easily reduced, there are numerous barriers to reform. Some of these include a perceived lack of safer alternatives, concerns about continuity of care, trainee education, and doctors’ preferences. As such, working hour reorganisation has been contentious globally. South Africa, a middle-income country where extended shifts are unregulated for most doctors, offers a useful case study of reform efforts. The South African Safe Working Hours campaign has promoted working hour reorganization through multi-level advocacy efforts, although extended shifts remain common. We propose that extended shifts should be regarded as an occupational hazard under health and safety legislation. We suggest options for managing the risks of extended shifts by adapting the hierarchy of controls for occupational hazards. Despite the challenges reform pose, the practice of unregulated extended shifts should not continue.

## 1. Introduction

Sleep deprivation, long accepted as an inseparable part of medical practice globally, is increasingly being recognized as a risk factor for patient and practitioner harm [1,2,3]. Rather than a weakness, the human need for regular sleep should be seen as a physiological reality. This poses a problem in healthcare, where qualified staff must be available at all hours of the day.

This article focuses on the after-hours work of doctors (although it could be applied to any health worker working extended shifts), which can be referred to as being “on-call” [4], extended duration work shifts [5], extended shifts [2], or continuous duty [6]. Whereas some disciplines allow doctors to work remotely on standby, this article focuses on the work of doctors who are required to be on-site for an extended period. We will use the term extended shifts throughout this article, defined as working more than 16 h continuously.

We recognize that there is variation in the opportunity for rest during an extended shift, depending on the discipline, workload, and seniority. In many countries there is a dearth of regulations pertaining to the maximum duration of an extended shift and it should be of concern that doctors have been found to work continuously with little or no rest for up to 76 h [7] (p. 9). While other work features that contribute to fatigue include frequency of shifts, total number of hours worked, and intensity and complexity of tasks, we limit our discussion to extended shifts.

We suggest that extended shifts and the resulting fatigue are an occupational risk factor (by increasing the risk of other workplace injuries) and an occupational hazard in themselves. We argue that specific regulation is required to mitigate this hazard whilst balancing the safety of practitioners and patients. In some countries this is enabled by existing occupational health laws. When applied to the risk posed by fatigue in medical settings, these laws oblige employers to take steps to mitigate this risk. We will present a case study from South Africa, illustrating efforts undertaken to achieve such risk mitigation regarding doctors’ working hours. Through the case study we will explore the contextual factors which may have bearing on such attempts.

## 2. Origins of Extended Shifts in Medical Practice

The on-call system utilized by medical doctors globally has been part of hospital-based healthcare for over a century. Although it is likely that this system has developed differently in many regions of the world, the USA is an influential example. In the late 19th century, pioneering surgeon, William Halsted, introduced multiple innovations in surgical procedures, including the use of gloves during surgery, and rigorous training for prospective surgeons [8]. He expanded the apprenticeship for surgeons through a period of “residency”—which involved staying on the hospital site and being available for emergencies, at all hours of the day and every day of the year [9]. This “restricted lifestyle” also meant residents received little or no pay, and were discouraged from marriage [9]. Halsted introduced the use of cocaine for topical anaesthesia, but also sustained his own “excellent physical vigour” by abusing this stimulant drug [8,10].

## 3. The Impact of Extended Shifts on Safety

A great deal has been learned about workplace and patient safety since Halsted’s residency programme was instituted. A 2011 systematic review that included studies of health care and non-medical industries concluded that both shift work and long working hours had a ‘substantial and well-documented detrimental effect on safety’ [11]. The review reports increased accident rates, with the findings most relevant to safety-critical activities such as transport and health sectors. It also found that work periods exceeding 8 h carry an accumulating risk of accidents such that the risk of accidents at around 12 h is twice the risk at 8 h. Extended shifts are strictly regulated in aviation, road freight, and passenger-carrying vehicle industries, yet in many countries the medical industry lags behind [12] (p. 2).

The death of Libby Zion in New York in 1984 as a result of a fatigue-related error made by a junior doctor is often cited as a turning point in working hour organisation and patient safety in the USA. An enquiry into contributing factors to her death led to limitation of the maximum duration of an extended shift in the US for residents to 24 h [9,13].

Extended shifts may lead to patient harm through medical errors [2]. In a prospective, randomized study of junior doctors in the United States, those working extended shifts (>24 h) compared to shorter shifts (<16 h) made substantially more serious medication errors (20.8%), and 5.6 times as many serious diagnostic errors [14]. Fatigue may also affect empathy and patient interaction negatively [15] (p. 11).

There is risk of acute harm to practitioners. A systematic review demonstrated an elevated risk for road traffic incidents in doctors after extended shift work [16]. The largest study included in the review followed 2500 junior doctors in the USA over a year and found that those working an extended duration shift (>24 h) were 2.3 times more likely to have a motor vehicle car crash, and 5.9 times more likely to have a near miss, than following a non-extended duration shift [17]. A prospective cohort study of 2737 intern doctors in the USA showed percutaneous injuries were more frequent during extended shifts than non-extended shifts [18].

## 4. Regulating Extended Shifts

There are global examples of profession-wide regulations aimed at patient and practitioner safety. The European Working Time Directive places limits on both continuous and total hours worked [3] (p. 86). It stipulates a minimum rest period per day of 11 h, effectively limiting the maximum duration of an extended shift to 13 h. In Australia, there is a National Code of Practice—Hours of Work, Shiftwork and Rostering for Hospital Doctors—released in 1999. The Australian Medical Association utilizes a Risk Assessment Guide and a Risk Assessment Checklist instead of absolute limits on working hours to help assess the risk level of an individual’s working hours [7].

In the United States in 2011 the Accreditation Council for Graduate Medical Education (ACGME) introduced a 16-h limit for extended duration shifts for first-year residents. A randomized control trial comparing standard 16 h shifts with shifts exceeding 16 h, found noninferior patient outcomes and no significant difference in residents’ satisfaction with overall well-being and education quality [19]. This study was used as justification in rolling back maximum working hour limits under ACGME regulations, despite the authors’ conclusion being contested [20].

## 5. Barriers to Change

Each of the above instances of regulation and reform has met, or continues to receive, considerable resistance and we discuss some of the salient objections to limiting extended shifts.

### 5.1. Shortage of Doctors

There is a global shortage of health workers (including doctors) which is not projected to be resolved in the near future without extensive intervention [21] (p. 44). Using this objection in favour of extended shifts presumes regulation must inevitably entail shortening overall number of hours (which has been the case in Europe and the US). We contest this as well as the assumption that extended shifts make the best use of the existing scarce pool of doctors. To the contrary, extended shifts may push practitioners from the profession or towards working in certain specialities where extended shifts are less common [22]. As outlined below, risk mitigation does not necessarily have to rely on having more doctors or reduced overall hours.

### 5.2. The Hazards of Shift Work

There are numerous ways to reduce fatigue whilst still providing after hours care. These include limiting consecutive hours worked by doctors by instituting non-extended shifts (lasting less than 16 h, hereafter “shift work”) and measures such as pre-call rest periods, floating shifts, and overlapping shifts [23].

Shift work can refer to a variety of shift lengths [3] (p. 74) or to work outside of the hours of 09h00–17h00 [24]. Nurses, for example, commonly work 12 h shifts [25]. Shift work is associated with an increased risk of accidents, especially on the night shift and increasingly as the shift exceeds 8 h [26]. Furthermore, shift work is associated with well-studied health hazards including increased risk of cancer [27], cardiovascular disease [28], and metabolic disorders [29]. It is not clear weather the harmful sequelae of shift work result from circadian disruption, disturbed sleep, associated behaviours, psychological stress, or an interaction between these and other proposed mechanisms [29]. At least some of the harm demonstrates a dose-response relationship with night shift work. Therefore a move to working shifts would not eliminate practitioner harm. However, it is likely that extended shifts include many of the risks of shorter shift work whilst further increasing the risks of errors or accidents as the duration of the shift increases.

Shorter shifts necessarily mean that doctors hand over patient care more frequently to the incoming team. There is some concern that this increases the risk of error and detracts from continuity of care. It is worth noting that a systematic review on this topic did not find reduced quality of care with shifts less than 16 h [30]. If patient care is the chief priority it is counterintuitive to present extended shifts as a solution, as these have been associated with increased treatment and diagnostic error [14].

### 5.3. Concerns about Education

Opinions differ on the impact of limiting extended shifts on medical education. Some fear this would reduce the overall quality of medical education due to lack of clinical exposure. A 2010 review found that a maximum shift duration of 16 h had no adverse effect or possibly a positive effect on the educational outcomes of medical residents [30].

Both acute and chronic sleep deprivation are known to impair memory among doctors [3] (p. 85). Although shorter shifts would not necessarily reduce chronic sleep deprivation, it is difficult to argue that extended shifts contribute to more learning, when much of the time worked is spent in a state of profound fatigue.

There are also concerns that shortened shifts will prevent acclimatisation to fatigue. However, repeated extended shifts have not been shown to decrease the negative impact of fatigue through acclimatisation [1,6,31].

### 5.4. Doctors’ Preference

Doctors may prefer to work extended shifts or that their subordinates do so, as this allows for more uninterrupted time off [3] (p. 74). In any other safety critical discipline, such as aviation, preference for a dangerous arrangement would not be considered when developing regulations. Promoting greater awareness among doctors about the risks of extended shifts to patients and themselves could play an important role, although this would not obviate the need for regulation.

## 6. Case Study: The Safe Working Hours Campaign in South Africa

In South Africa, medical doctors provide after-hours care by working 24 h shifts where they are “on-call” on site, followed by a day of “post-call” work [4] (p. 62). Although rest may be possible, there is usually no regulation to prevent a doctor from working continuously throughout this period, sometimes up to 36 h [4] (p, 63). This contrasts with nurses, allied health professionals, and laboratory workers, who are part of the 24 h provision of healthcare but provide this care in 8 or 12 h shifts [4] (p. 52)

During their two-year internship after completing medical school, South African junior doctors receive placements in training facilities across the country, rendering after hours services in speciality departments such as surgery, obstetrics and gynaecology, and internal medicine. They receive varying degrees of support, from on-site supervision to telephone-based consultations with seniors and perform procedures that include caesarean sections and trauma care.

Doctors’ legal rights to demand working hour protections offered by the Basic Conditions of Employment Act of 1997 are curtailed by their earning more than the specified threshold of R205 433.30 (approximately €9 800) per annum [32]. The right to “demand” is replaced by the right to “negotiate” specific overtime arrangements with employers. However, threshold exemptions do not absolve the employer of the need to comply with occupational health laws [33].

Prior to 2017, the Health Professions’ Council of South Africa (HPCSA), the regulatory body for medical professionals, had set a 30-h limit on extended shifts for medical graduates during their two years of internship [34]. Beyond these first two years of work as a doctor, no maximum duration has been set.

The Safe Working Hours (SWH) campaign was started in 2014 by medical students and junior doctors, including four of the authors (K.K., H.v.d.W., E.v.L. and F.J.). The narrative account of this case study is informed by their experiences and publicly available information. The aim of the campaign was to generate an evidence-based discussion around working hour regulations in South Africa and advocate for revised regulations. An online petition asking for an urgent revision of doctors’ working hour regulations gathered over 7 000 digital signatures. SWH questioned the 30-h limit for intern doctors proposed by the HPCSA, calling for shift duration to be limited to 16 h. SWH volunteers consulted with various stakeholders to advocate for the introduction of maximum shift durations for medical doctors. These included the South African Medical Association (SAMA), the HPCSA, Western Cape Department of Health, National Department of Health and SWH representatives at hospitals throughout the country.

The junior doctor interest group of SAMA, Judasa, led a colour coded armband campaign to raise awareness among practitioners and patients of the harms of extended shifts. Doctors wore green armbands for the first 24 h of their shift, then switched to an orange armband, and from 30 h used a red armband. This campaign did not have sustained uptake. The reasons for this may include that it competed with other clinical priorities, did not directly engage those with the authority to change working conditions, or that some doctors felt their practice became unsafe during the ‘green’ period, i.e., before 24 h.

In 2016 the campaign received renewed attention after a junior doctor died in a motor vehicle accident while driving home after a 24-h extended shift. SWH created the ‘*Share your story*’ campaign, where doctors could share their experiences of extended shifts. These included adverse effects on patient care, multiple reports of motor vehicle accidents while driving home after extended shifts, and the psychological impact of working while sleep impaired. A video compilation of the ‘*Share your story*’ campaign reached approximately 17,000 viewers on social media. A reuploaded version is available online [35] (see Figure 1). The SWH campaign was featured in 23 national newspaper articles, 17 radio interviews and two opinion pieces by campaign volunteers for national newspapers and on national television. *Township ER* [36], a documentary that follows five South African junior doctors on an extended shift, was screened at various health facilities and medical schools to facilitate discussion.

In 2016, two years after the SWH campaign started, the HPCSA circulated a document outlining changes to internship training regulation, shortening maximum allowable shift duration from 30 h to 24 h with a two-hour period for patient hand-over. This was included in a 2017 guideline on internship training [37]. It is not yet clear whether the 24 h regulation has been implemented throughout South Africa, although an advocacy report published on the SWH website suggests that extended shifts are still practiced widely [38].

The HPCSA’s maximum shift duration for interns demonstrates an important acknowledgement of the potential for harm to patient and practitioners. However, it is doubtful whether a 24 h limit is sufficient to mitigate risk of harm. The shift limit contrasts sharply with the unregulated shifts which doctors can be expected to work beyond internship.

The National Department of Health also formed a Safe Working Hours task team in 2016, but this does not appear to be active. The campaign was successful in engaging with the public about the importance of working hour regulations and amplifying the voices of junior doctors who bear the brunt of extended shifts. Some senior doctors voiced opposition to the campaign, deeming extended shifts a rite of passage and expressing reluctance to be involved in additional handover rounds associated with shorter shifts.

An alternative route would be to use existing occupational health legislation in South Africa. Given the risks of extended shifts described above, it can be argued that this practice contravenes the country’s Occupational Health and Safety Act of 1993. This law states that it is the employer’s responsibility to provide a safe working environment and to take steps to mitigate risks to both employees and others “directly affected” [39]. Medical employers would have to demonstrate that they have undertaken the risk assessment required by the Act

We believe that given the probability that extended shifts may result in harm (summarised above), employers would have to take steps to mitigate this risk (See Table 1). The South African National Health Act of 2003 also obligates the employer to minimise injury to health care personnel and this poses another avenue to hold employers accountable [40]. To our knowledge, these interpretations have not yet been tested in South African courts with regards to extended shifts.

## 7. Extended Shifts as an Occupational Hazard

In taking steps to mitigate the hazard of extended shifts, it is useful to apply the hierarchy of controls, such as the one described by the US National Institute of Occupational Safety and Health [41] (See Figure 2). Usually used for physical, dust, and chemical hazards, the hierarchy arranges solutions in order of theoretical effectiveness based on the degree of control of the hazard. Table 1 demonstrates how this hierarchy may be adapted to extended shifts. Conceiving of extended shifts in this manner has the purpose of categorising appropriate and feasible steps to address the hazard in a stepwise manner within an occupational health framework.

**Table 1 ijerph-17-05853-t001:** Adapting the hierarchy of controls to extended shifts.

Control	Examples as Applied to Extended Shifts
**Elimination**	Instituting a cap on shift durations
**Substitution**	Moving to one of a variety of *alternative shift systems*, within the safest acceptable parameters: Staggered shifts, float systems [23] Pre-call naps [11] Guaranteed periods of rest on longer shifts [43] Minimum intervals between shifts [44] No more than 4 consecutive night shifts [44]
**Environmental Measures**	Providing rest facilities and ensuring that they are conducive to rest [37] Ensuring adequate light in the working environment in order to promote wakefulness [3] (p. 76)
**Administrative Measures**	Ensuring that employment contracts allow for a change from extended shifts to shorter shifts. (For example, not insisting that overtime can only be worked during certain hours of the day) * Providing schedulers assistance in drafting more complex after-hours schedules [23] Coupling the above changes with interventions to minimize the harm from shifts to patients and practitioners: Hand-over checklists, including digital records and alert systems [45] Education about the hazards of shift work and how to mitigate these [3] (p. 75)
**Personal Protective Measures**	Preventing the increased risk posed to patients by those on extended shifts by: “Buddy systems”, where a rested person checks the work of the post-call person * System of additional checks for high risk work [46] Signalling which doctors are most fatigued through colour coded armbands [47] Preventing death or injury of the practitioner: Delegating dangerous procedures * Encouraging the use of public transport [46]

* Suggested examples obtained from practitioners and managers during the Safe Working Hours campaign in South Africa.

The measures listed are not exhaustive and consist of suggested examples obtained from published literature and regional guidelines, where cited, as well as from practitioners and managers during the Safe Working Hours campaign in South Africa (marked with *). Research as well as trial-and-error implementation are needed to determine the effectiveness of the various measures. We have also attempted to reflect the role of individual doctor agency by placement lower down within the hierarchy.

## 8. Conclusions

The medical profession has become inured to the risks associated with the historic practice of extended shifts, and has not modified this practice to reflect emerging evidence. Conceiving of extended shifts as an occupational hazard brings this danger into focus and into the domain of occupational health. It also directs attention to a menu of options to mitigate potential for harm. This approach should make risk reduction possible despite a shortage of doctors, and without necessarily reducing total hours worked. This assessment can then be balanced against considerations such as continuity of care and safe handover. Working at night or in shifts will always entail some increased health risk. However, leaving extended shifts unregulated in any country should be regarded as unacceptable.

The South African case study shows that in addressing the problem of extended shifts in any jurisdiction, context matters. It illustrates how different advocacy strategies have been used for specific regulatory reform, with limited success. The case study also shows how existing labour and occupational health legislation could be applied to the hazard of extended shifts. Medical practitioners must recognise that extended shifts are not an immovable or intrinsic part of healthcare practice, but an occupational hazard which must be addressed.

## Figures and Tables

**Figure 1 ijerph-17-05853-f001:**
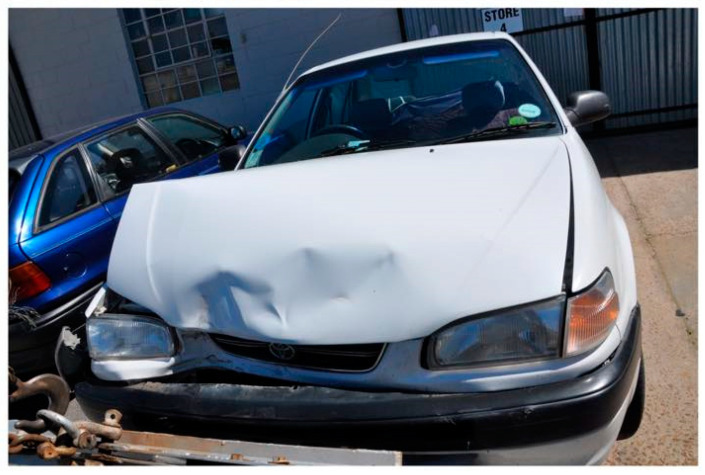
A photo of a motor vehicle crash submitted by a junior doctor as part of the ‘*Share your Story*’ campaign. The intern doctor worked a 26-h extended shift and was required to drive to a clinic to work an additional 5 h, when they were involved in a collision. Reproduced with permission from the photographer.

**Figure 2 ijerph-17-05853-f002:**
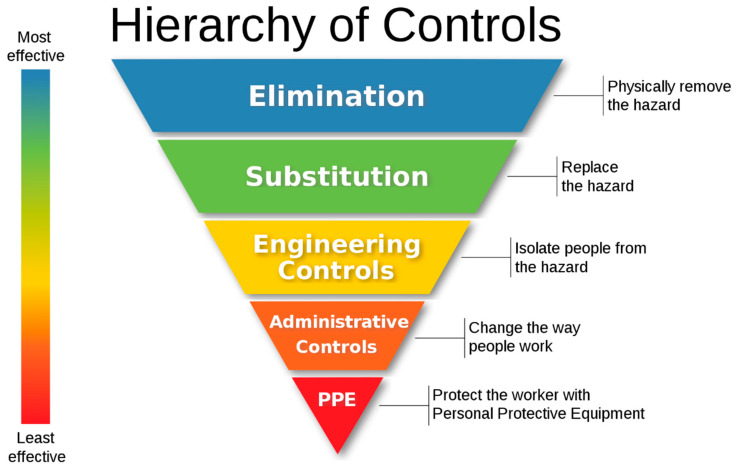
The hierarchy of controls for occupational hazards. Copyright: public domain [42].

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
