# Peer review of "Doctors’ Extended Shifts as Risk to Practitioner and Patient: South Africa as a Case Study"

_ijerph, 2020, doi:10.3390/ijerph17165853_

Round 1

Reviewer 1 Report

Dear authors,

This paper describes the campaign for introducing the policy to regulate extended shift or medical practitioners, which is interesting and well-written. I have some recommendation for you as following:

  1. The reference #39 refers the armbands to show doctors’ fatigue level, which seems to be interesting and effective methods. This is briefly mentioned in the Table 1, but it is worth to describe in detail. I recommend you’d describe the armbands for doctors in the text.

  1. The HPCSA have shortened the maximum shift duration from 30 hours to 24 hours, but I think it is not enough to prevent the health hazards and medical errors of medical practitioners. I think you should describe the future plans for extended shifts, if possible.

  1. I think the methods of control in Table 1 should coincide with Figure 1. (eg. Environmental control à engineering control)

Author Response

Thank you for your feedback on our submitted manuscript

  1. We agree that this intervention is interesting.It was promoted during the period of the SWH campaign in South Africa. We have included a paragraph in the text to discuss how this was implemented at line 213. The arm bands stemmed from an initiative by the South African Medical Association in 2016, following heightened public interest into the matter, but has not been maintained. The armbands represent an interesting means of raising awareness and perhaps personal protection. The colour coding that was used were 0-24hours on duty(green), 24-30hours(orange) and more than 30 hours (red). These limits received some criticism as they were not evidence-based. There are also other considerations, such as whom the arm bands are meant to signal fatigue to (the practitioner themselves, patients, or colleagues) or whether armbands absolve the employer of a responsibility for more effective interventions.
  2. On consideration, we agree that we should have made this more explicit. We have inserted an additional section from lines 242 to 308in order to address this and clarified that we did not view the 24 hour shift limit adequate or based on evidence.We suggest that the most rational approach to this problem would entail introducing shift limits which are more in line with some of the existing literature as well as expanding regulation to all medical practitioners who provide on-site after hours care. This will inevitably pose several challenges, requiring the redesign of overtime contracts and shift systems. This would also require that doctors make a clear distinction between those who are truly on-call and those who are providing bedside care after hours, as well as how grey this distinction can become (for example, when a radiologist has an unusually busy night of reading scans, but is still required at work the next day). We believe that an occupational hygiene approach, as outlined in section 7 of our manuscript offers a valuable starting point for this analysis, and will assist administrators and practitioners to balance risks and select the safest route under the constraints posed.
  3. Whilst we recognize the need for concordance between the figure and the table, we believe that the adaptation in terms of the category names is necessary in order to apply to working hours. As such, we have had to adapt category names on the hierarchy. As this is an adaptation,rather than a replication, we hope that you will agree that the changes in table 1 accomplish this goal, whilst remaining true to the meaning of the hierarchy of controls. (Line 338) We are grateful for the opportunity to improve our manuscript in light of the reviewersinput and hope that our discussion here reflects the efforts we have made to act upon their feedback.

Reviewer 2 Report

I want to thank you for the opportunity to review this manuscript. The time spent creating and shipping it is greatly appreciated. The document makes an interesting analysis of the risks associated with the practice of extended shifts among medical professionals, finally commenting on the specific situation in South Africa in this regard. However, its presentation as a comment limits the contributions that the work makes to the field of scientific research on the subject.

Thus, I consider that, despite the fact that the subject is interesting and the document is correctly written, it should be focus on research to contemplate its possible publication in the journal. Since, currently, it is still an informative document.

I encourage the authors to continue this interesting line of research and to improve the quality of the manuscript, in order to achieve the appropriate research standards to qualify for publication in scientific journal.

Author Response

We would like to thank Reviewer 2 for their feedback with regards to our manuscript. Our manuscript is not original research, but a commentary.This includes an illustrative case study, evaluation and synthesis of available research, and the application of existing legal and ethical frameworks, and provides what we believe is a novel perspective on the contentious field of working hour research in medicine.Given the paucity of similar syntheses (particularly focussed on low-and middle income countries), we believe that our commentary adds an important set of ideas to this field and would encourage reviewers to consider this in their ongoing evaluation of the manuscript.

Reviewer 3 Report

Dear Authors I carefully evaluated the study, finding it overall well written and well presented. The commentary is interesting and there is a need to investigate these aspects.

I suggest to improve literature search in order to reinforce some concepts only partially explained. In detail, the effects of shift work, in the paragraph “The hazards of shift work” should be deepened.

I can not find any reference supporting the sentence “Furthermore, shift work is associated with well-studied health hazards including increased risk of cancer, cardiovascular disease, metabolic and reproductive disorders. This part should be improved, also considering some relevant references about the health effects of shift work.

Authors can also refer to these articles: Kecklund G, Axelsson J. Health consequences of shift work and insufficient sleep. BMJ. 2016;355:i5210. Published 2016 Nov 1. doi:10.1136/bmj.i5210 Lecca LI, Setzu D, Del Rio A, Campagna M, Cocco P, Meloni M. Indexes of cardiac autonomic profile detected with short term Holter ECG in health care shift workers: a cross sectional study. Med Lav. 2019;110(6):437-445. doi:10.23749/mdl.v110i6.8048 Torquati L, Mielke GI, Brown WJ, Kolbe-Alexander T. Shift work and the risk of cardiovascular disease. A systematic review and meta-analysis including dose-response relationship. Scand J Work Environ Health. 2018;44(3):229-238. doi:10.5271/sjweh.3700 Reynolds AC, Paterson JL, Ferguson SA, Stanley D, Wright KP Jr, Dawson D. The shift work and health research agenda: Considering changes in gut microbiota as a pathway linking shift work, sleep loss and circadian misalignment, and metabolic disease. Sleep Med Rev. 2017;34:3-9. doi:10.1016/j.smrv.2016.06.009

Author Response

Thank you for the valuable feedback you have provided with regards to providing more depth in our article. We have made updates to our references (lines 133-143) and have added additional clarifications with regards to shift work and added additional explanations about the possible mechanism of harm. Furthermore, we have revisited the evidence with regards to reproductive disorders and, as the evidence of harm is not as well established as with bowel or breast cancer, cardiovascular and metabolic disease, have decided to omit this from the list, which is intended to be illustrative and not exhaustive.Our article’s central premise is that extended shifts should not be unregulated, and thus need an occupational hygiene approach to risk mitigation. We believe that a future and as yet unestablished,area of research in medicine will entail the careful weighing of risks and benefits in order to determine the safest work schedules for patients and practitioners, depending on the discipline, workforce and patient numbers. We are grateful for the opportunity to improve our manuscript in light of the reviewersinput and hope that our discussion here reflects the efforts we have made to act upon their feedback.

Round 2

Reviewer 2 Report

Dear authors,

Regretfully, I consider that this type of document is not appropriate for a research journal, so I reaffirm my previous decision. I want to thank you for your time and, as I already mentioned, I encourage you to transfer your idea to the empirical field.

Receive a warm greeting.

Author Response

We have carefully considered and discussed the comments made by Reviewer  on our manuscript:
Round 1:

I want to thank you for the opportunity to review this manuscript. The time spent creating and shipping it is greatly appreciated. The document makes an interesting analysis of the risks associated with the practice of extended shifts among medical professionals, finally commenting on the specific situation in South Africa in this regard. However, its presentation as a comment limits the
contributions that the work makes to the field of scientific research on the subject.
Thus, I consider that, despite the fact that the subject is interesting and the document is correctly written, it should be focus on research to contemplate its possible publication in the journal. Since, currently, it is still an informative document.
I encourage the authors to continue this interesting line of research and to improve the quality of the manuscript, in order to achieve the appropriate research standards to qualify for publication in scientific journal.

Round 2:

Dear authors,
Regretfully, I consider that this type of document is not appropriate for a research journal, so I reaffirm my previous decision. I want to thank you for your time and, as I already mentioned, I encourage you to transfer your idea to the empirical field.
Receive a warm greeting.

Author responses:

Whilst we acknowledge the reviewer’s high valuation of the need for empiric research, we respectfully disagree with the view that a commentary offers limited contributions to the field of scientific research on the subject.

Commentaries are a widely recognized format for scholarly publications. One example, published in this journal, includes:

Tong, M. X.; Hansen, A.; Hanson-Easey, S.; Cameron, S.; Xiang, J.; Liu, Q.; Sun, Y.; Weinstein,P.; Han, G. S.; Williams, C.; et al. Infectious Diseases, Urbanization and Climate Change: Challenges in Future China [1].

The IJREPH also publishes “Pespectives”, which have a similar format to commentaries, such as:

Woolfenden, S.; Milner, K.; Tora, K.; Naulumatua, K.; Mataika, R.; Smith, F.; Lingam, R.; Kado, J.; Tuibeqa, I. Strengthening Health Systems to Support Children with Neurodevelopmental Disabilities in Fiji—A Commentary [2]

These examples summarise and comment on the existing literature, whilst integrating disparate strands of research, such as urbanization, climate change and emerging infectious diseases.

The value of commentary articles is that they offer an in-depth discussion of a study (in this case multiple scholarly articles and a case study), as well as aiding readers in incorporating this new knowledge into practice and stimulating further research. [3]

Our commentary article integrates research from numerous fields (such as sleep medicine, occupational health and labour legislation) and offers an analysis of working hour reform in a middle-income country, as and the implications of the latest research findings for contemporary practice in this and similar settings.
We are grateful for the opportunity to respond to the reviewer’s comments but wish the article to be evaluated as revised for publication as a Perspective piece in the special edition of the Journal, if the editors believe this to be the appropriate category.

References

[1] Tong, M. X.; Hansen, A.; Hanson-Easey, S.; Cameron, S.; Xiang, J.; Liu, Q.; Sun, Y.; Weinstein, P.; Han, G. S.; Williams, C.; et al. Infectious Diseases, Urbanization and Climate Change: Challenges in Future China. Int. J. Environ. Res. Public Health, 2015, 12 (9),11025–11036. https://doi.org/10.3390/ijerph120911025.

[2] Woolfenden, S.; Milner, K.; Tora, K.; Naulumatua, K.; Mataika, R.; Smith, F.; Lingam, R.; Kado, J.; Tuibeqa, I. Strengthening Health Systems to Support Children with Neurodevelopmental Disabilities in Fiji—A Commentary. Int. J. Environ. Res. Public Health, 2020, 17 (3), 1–11. https://doi.org/10.3390/ijerph17030972.

[3] Jull, A. How to Write a Commentary-an Editor’s Perspective. Evid. Based. Nurs., 2007, 10(4), 100–103. https://doi.org/10.1136/ebn.10.4.100.
